# Determinants of Allergic Sensitization, Asthma and Lung Function: Results from a Cross-Sectional Study in Italian Schoolchildren

**DOI:** 10.3390/ijerph17145087

**Published:** 2020-07-14

**Authors:** Gaspare Drago, Silvia Ruggieri, Giuseppina Cuttitta, Stefania La Grutta, Giuliana Ferrante, Fabio Cibella

**Affiliations:** 1National Research Council of Italy, Institute for Biomedical Research and Innovation, 90146 Palermo, Italy; gaspare.drago@irib.cnr.it (G.D.); silvia.ruggieri@irib.cnr.it (S.R.); giuseppina.cuttitta@irib.cnr.it (G.C.); stefania.lagrutta@irib.cnr.it (S.L.G.); 2Department of Health Promotion, Mother and Child Care, Internal Medicine and Medical Specialties, University of Palermo, 90100 Palermo, Italy; giulianafer@hotmail.com

**Keywords:** cigarette smoking, child health, acute respiratory infections, allergic sensitization, asthma

## Abstract

Prenatal smoking exposure and early-life respiratory infections are major determinants of asthma during childhood. We investigate the factors influencing allergic sensitization (AS), asthma, and lung function in children and the balance between individual and environmental characteristics at different life stages. 1714 children aged 7–16 years and living in southern Italy were investigated using a parental questionnaire, skin prick tests, and spirometry. We found 41.0% AS prevalence: among children without parental history of asthma, male sex, maternal smoking during pregnancy (MatSmoke), and acute respiratory diseases in the first two years of life (ARD2Y) were significant risk factors for AS. MatSmoke was associated (OR = 1.79) with ARD2Y, and this association was influenced by sex. ARD2Y was, in turn, a significant risk factor (OR = 8.53) for childhood current asthma, along with AS (OR up to 3.03) and rhinoconjuctivitis (OR = 3.59). Forced mid-expiratory flow (FEF_25–75%_) was negatively affected by ARD2Y, with a sex-related effect. Thus, males exposed to MatSmoke had significantly lower FEF_25–75%_ than unexposed males. Despite the difficulty of discriminating among the complex interactions underlying the development of allergic respiratory diseases, ARD2Y appears to strongly influence both asthma and lung function during childhood. In turn, ARD2Y is influenced by prenatal exposure to tobacco smoke with a sex-dependent effect.

## 1. Introduction

Asthma is the most common chronic childhood disease [1], and large variations in the prevalence of asthma and rhinitis symptoms in children have been reported worldwide [2]. In Italy, the SIDRIA (Italian Studies on Respiratory Disorders in Children and the Environment) Study, the largest Italian epidemiological survey of children, on the basis of questionnaire responses, reported a prevalence of 10.4% of lifetime asthma and 17.4% of rhinoconjunctivitis in the previous 12 months in 10,267 Italian adolescents [3], suggesting a trend towards a stabilization of asthma prevalence.

Asthma is a common, heterogeneous, chronic respiratory disease affected by genetic, socioeconomic, and environmental factors [4]. Allergic sensitization is highly associated with the risk of respiratory symptoms among asthmatic children [5], and atopy increases the probability of allergic asthma [4]. Among environmental factors, pre- and postnatal passive smoking exposure [6,7] along with early life respiratory infections [8] constitute the major determinants of asthma and poorer lung function during childhood.

The present work investigates the factors influencing allergic sensitization, asthma, and lung function in children and the balance between individual characteristics and environmental exposures at different life stages: prenatal, early life, and childhood. To do this, we pooled data from two previous cross-sectional population studies performed in the Mediterranean area of southern Italy, which investigated respiratory and allergic symptoms, also objectively evaluating the respiratory function and the sensitization to most common aeroallergens.

## 2. Materials and Methods

### 2.1. Study Population

A total of 2493 Caucasian Sicilian schoolchildren (primary and lower secondary school), aged 7–16 years were pooled from two previous cross-sectional studies with nearly identical study designs: (i) Cibella et al. [9], *n* = 2150 from 16 schools (November 2005–May 2006); (ii) *CCM-Indoor school* project [10], *n* = 343 children from 8 schools (April 2012–April 2013).

### 2.2. Questionnaires

Parents of all 2493 investigated children completed a modified ISAAC (International Study of Asthma and Allergies in Childhood) questionnaire [11] regarding personal information and questions about past and current respiratory allergic symptoms and possible exposures. In both studies, the same core questions were used. The complete list of questions used in the analysis is the following:Parental history of asthma *Is there any case of asthma in the family? (mother or father)*Maternal smoking during pregnancy *Did the mother smoke during pregnancy? (Yes/No)*Paternal smoking during pregnancy *Did the father or other family members smoke in presence of the mother during her pregnancy? (Yes/No)*Early- life (during the first two years of life) exposure to environmental tobacco smoke (Early ETS) *During the first two year of life, was your child exposed to tobacco smoke? (Yes/No)*Acute respiratory diseases during the first two years of life *During the first two years of life, did your child suffer from any infections, such as pneumonia, bronchitis, asthmatic bronchitis, bronchiolitis? (Yes/No)*Wheeze in the last 12 months *In the past 12 months, has your child had wheezing or whistling in the chest? (Yes/No)*Doctor diagnosed asthma *Has your child ever had asthma diagnosed by a doctor? (Yes/No)*Nocturnal cough *In the past 12 months, has your child had a dry cough at night, apart from a cough associated with a cold or chest infection? (Yes/No)*Exercise wheeze *In the past 12 months, has your child’s chest sounded wheezy during or after exercise? (Yes/No)*Current exposure to tobacco smoke at home (ETS) *Is your child exposed to tobacco smoke in the dwelling? (Yes if daily or often [1–4 times/week])*Presence of mould/dampness at home *Is there dampness or visible mould growth in your child’s bedroom? (Yes/No)*Pet ownership *Does the child, or any other family member, have any pets in the dwelling? (Yes/No)*Presence of heavy vehicular traffic in proximity to home *Does your child live within 200 m of a street with heavy traffic? (Yes/No)*Rhinoconjunctivitis *In the past 12 months, has your child had a problem with sneezing, or a runny or blocked nose when he/she did not have a cold or the flu, with this nasal problem being accompanied by itchy-watery eyes? (Yes/No)*

Moreover:Current asthma was defined as a positive answer to both the questions relevant to Wheeze in the last 12 months and Doctor diagnosed asthma.“Healthy” children were selected as those individuals without Doctor diagnosed asthma or without wheeze, nocturnal cough, exercise wheeze, or rhinoconjunctivitis in the last 12 months.

### 2.3. Skin Prick Tests

Skin prick tests were performed according to EAACI (European Academy of Allergy and Clinical Immunology) recommendations [12] with a standard panel including *Dermatophagoides* mix, *Alternaria*, *Parietaria judaica*, grass mix, *Olea*, *Blattella germanica*, dog and cat dander, plus a positive (histamine 1%) and a negative (saline) control (Stallergènes Italia S.r.l., Milan, Italy). The reading was performed after 15 min: reactions were considered positive if the mean wheal diameter (computed as the maximum diameter plus its orthogonal divided by 2) was 3 mm or greater, after having subtracted the wheal diameter of the reaction to the negative control. Allergic sensitization was defined as the presence of at least one positive skin prick test. The atopic index (AI) was computed as the number of the individual positive skin prick tests and divided into four classes: 0-non atopic, 1-one positive skin test, 2-two positive skin tests, 3-three or more positive skin tests.

### 2.4. Spirometry

Height (in cm) and weight (in kg) were measured for all the children in standing position without shoes using a stadiometer and an electronic digital scale. BMI was computed as weight/height^2^ (kg/m^2^). Pulmonary function tests were performed with a portable spirometer (MicroLoop, Micro Medical, Chatham Maritime, Kent, UK). Forced expiratory volume in one second (FEV_1_), forced vital capacity (FVC) and maximum mid-expiratory flow (FEF_25–75%_) were measured according to ATS/ERS (American Thoracic Society/European Respiratory Society) guidelines [13]: the best FVC and FEV_1_ were retained and FEF_25–75%_ was selected from the manoeuver with the largest sum of FEV_1_ and FVC. For each subject, FEV_1_/FVC and FEF_25–75%_/FVC ratios were also computed. Spirometric predicted values were those from Quanjer et al. [14]. Spirometric data were evaluated as z-scores.

The studies were approved by the University Hospital of Palermo Ethics Committee on 8 January 2007 and 18 May 2011. Parents of the invited schoolchildren signed written informed consent. According to Italian law, respect of individual privacy concerning clinical data was guaranteed.

### 2.5. Statistics

The normality of distribution of the continuous variables was evaluated by the Kolmogorov-Smirnov Test. Continuous variables were expressed as means ± standard deviations (SD) and evaluated by one-way Analysis of Variance (ANOVA). Categorical variables were expressed as percentages and compared by means of the χ^2^ test. Multiple logistic regression models were used for investigating the effects of different variables on binary outcomes, taking into account confounders and effect modifying variables. These latter, were selected from the outdoor (residence in proximity to roads with intense vehicular traffic) and indoor (early and current exposure to environmental tobacco smoke, mould/dampness exposure at home, pet ownership) exposures known to possibly affect the evaluated outcome variables. Similarly, multiple linear regression models were created for investigating the effects of different variables on continuous outcomes. *p*-Values < 0.05 were considered significant.

Statistical analyses were performed by IBM SPSS statistics v20, IBM, New York, NY, USA.

## 3. Results

Out of 2493 total children, 1714 (F = 833, 48.6%) with complete data and parental questionnaires (68.9%) were evaluated. Descriptive and pulmonary function data of the population sample are shown in Table 1, for the overall sample and separately for subjects with and without current asthma. In our sample, 12.1% indicated maternal smoking during pregnancy, 31.9% reported the presence of acute respiratory diseases during the first two years of life (F/M: 29.2%/34.5%, *p* = 0.019), and 12.3% had positive parental asthma history.

In a multiple logistic association model, male sex, maternal smoking during pregnancy, and parental history of asthma were positively associated with acute respiratory diseases during the first two years of life, while no effect was found for paternal smoking during pregnancy and early ETS exposure (Table 2). When compared to females unexposed to maternal smoking during pregnancy, the odds ratio (OR) for acute respiratory diseases during the first two years of life in exposed females was 2.09 (95% CI 1.34/3.26, Table 3), while the OR of exposed males was 2.06 (95% CI 1.31/3.25).

Considering subjects without parental history of asthma (N = 1504), among children reporting acute respiratory diseases during the first two years of life, 44.7% presented allergic sensitization, while among children without acute respiratory diseases during the first two years of life, 37.2% (*p* = 0.007) showed allergic sensitization. Among subjects with parental history of asthma (N = 210), the same figures were 53.0% and 51.2% (*p* = 0.80). In a multiple logistic model for allergic sensitization, the factors male sex, parental history of asthma, and acute respiratory diseases during the first two years of life were positively associated with allergic sensitization during childhood (Table 4). Stratifying the same logistic model for parental history of asthma, none of the previous variables were associated with allergic sensitization among children with parental history of asthma (Table 5). Conversely, among children without parental history of asthma, male sex, maternal smoking during pregnancy, and acute respiratory diseases during the first two years of life were positively associated with allergic sensitization (i.e., AI).

In a separate multiple logistic model for current asthma, we tested the effect of previous (*in utero* and early-life) and current possible risk factors (Table 6): acute respiratory diseases during the first two years of life (OR = 8.53, 95% CI 4.15/17.52), rhinoconjunctivitis (OR = 3.59, 1.87/6.90), and allergic sensitization were positively associated with current asthma. In particular, AI class 1 produced an OR of 2.28 (95% CI 1.01/5.16), while class 3 was 3.03 (1.37/6.70). Consequently, the prevalence of current asthma in our sample progressively increased in the four AI classes, from 1.5% in class 0 up to 8.0% in class 3 (*p* for trend < 0.038). No effect was found between both current/past environmental exposures and current asthma.

In a multiple linear regression model for FEF_25–75%_ (as z-score), acute respiratory diseases during the first two years of life and current asthma were negatively correlated with FEF_25–75%_, and different combinations of sex and maternal smoking during pregnancy provided different results (Table 7): the model provided a B coefficient of −0.38 (95% CI −0.60/−0.15) for male sex and exposure to maternal smoking during pregnancy with respect to male children unexposed to maternal smoking during pregnancy, whereas exposed females showed a B coefficient of −0.24 (95% CI −0.46/−0.03). Consequently, the same four combinations of sex and exposure to maternal smoking during pregnancy produced significantly different levels of FEF_25–75%_ (z-score) between male children exposed or not to maternal smoking during pregnancy, but not among females (Figure 1). When stratifying the multiple linear regression for sex, any significant effect of exposure to maternal smoking during pregnancy among females disappeared (Appendix A). Moreover, it is noteworthy that, in the same model including only the 1102 “healthy” individuals, FEF_25–75%_ was still negatively correlated to both acute respiratory diseases during the first two years of life and to maternal smoking during pregnancy (Appendix A).

A similar model was built for FEF_25–75%_/FVC ratio (Appendix A): the different combinations of sex and exposure to Maternal smoking during pregnancy produced very similar B values for females, while males exposed to Maternal smoking during pregnancy presented a more negative B value (−0.189 [95% CI −0.247/−0.131]) than males unexposed to maternal smoking during pregnancy (−0.106 [95% CI −0.134/−0.079)].

## 4. Discussion

This cross-sectional study conducted on a sample of 1714 adolescents from the Mediterranean area of southern Italy was aimed at investigating factors influencing allergic sensitization, asthma, and lung function in children including individual characteristics and environmental exposures at different life stages (prenatal, early life, childhood).

We found that in utero exposure to maternal smoke produces an increased prevalence of acute respiratory disease during the first two years of life. Both exposure to maternal smoke during pregnancy and acute respiratory diseases during the first two years of life, in turn, affect–directly or indirectly–allergic sensitization, asthma prevalence, and lung function during adolescence, with a sex-dependent effect.

### 4.1. Acute Respiratory Diseases in the First Two Years of Life

We found that a parental history of asthma was a significant independent risk factor for acute respiratory diseases during the first two years of life [15], with atopy in synergy with recurrent early-life respiratory infections for higher prevalence of later asthma [16]. In the present work, sensitivity to Maternal smoking during pregnancy was similar between the sexes, although–among non exposed babies–the risk of acute respiratory diseases during the first two years of life was higher in males. Nevertheless, both males and females exposed to maternal smoking during pregnancy showed more than twice the risk (ORs 2.06 and 2.09, respectively) of having acute respiratory diseases during the first two years of life with respect to unexposed females, confirming the role of maternal smoking during pregnancy as a significant factor for increased risk of hospital admission for acute upper and lower respiratory infections, asthma, and bronchiolitis [17]. It has been suggested that maternal smoking during pregnancy negatively influences lung function in the newborn [18,19], while the effects on bronchial hyperresponsiveness remain uncertain [20,21]. Exposure to maternal smoking during pregnancy could also damage the epithelium impairing defense mechanisms and thus increasing susceptibility to pathogens [22]. We found an overall significant effect of sex on acute respiratory diseases during the first two years of life: this is probably due to the fact that, at birth, the lungs of boys are larger than those of girls [23], while maturation is more advanced in females. Thus, at birth, females have a more favorable airway/lung size ratio, while males present disproportionate growth (“dysanapsis”) between airways and air spaces [24], predisposing infants to developing bronchial obstruction during viral respiratory infections early in life [25].

### 4.2. Allergic Sensitization

We found that exposure to maternal smoking during pregnancy and male sex were significant risk factors for allergic sensitization during childhood. While the association between male sex and atopy up to puberty is well known [26,27], the relationship between maternal smoking during pregnancy and allergic sensitization is controversial. We found no significant effect of maternal smoking during pregnancy on allergic sensitization in children with parental history of asthma. Conversely, a previous work has reported that maternal smoking during pregnancy is a strong risk factor for allergic sensitization and asthma symptoms during the first 10 years of life, but only in children with allergic parents [28]; more recently, no clear associations were found between maternal smoking during pregnancy and atopic sensitization [29].

A further strong risk factor for allergic sensitization was acute respiratory diseases during the first two years of life. In this regard, the result obtained by stratifying children for parental history of asthma is interesting. Similarly to what was observed for maternal smoking during pregnancy, we found that among subjects without a parental history of asthma, acute respiratory diseases during the first two years were associated with a higher prevalence of allergic sensitization. Conversely, this result was not found in children reporting a positive family history of asthma. Thus, we suppose that the strong effect of parental history of asthma – when present – is able to hide the effects of acute respiratory diseases during the first two years of life on allergic sensitization. Despite the hypothesis that allergic sensitization increases the risk for early-life respiratory infections and not the inverse [15], our results suggest that acute respiratory diseases during the first two years of life produce their effect on future allergic sensitization mainly in subjects with apparently lower genetic predisposition (i.e., lack of parental history of asthma). However, it has to be acknowledged that the latter result could be influenced by the limited number (N = 210) of subjects who reported a parental history of asthma.

### 4.3. Current Asthma

In our study, current asthma was defined as the contemporary presence of doctor diagnosed asthma and wheeze in the last 12 months. In a multiple logistic model, acute respiratory diseases during the first two years of life, rhinoconjunctivitis and allergic sensitization were independent risk factors for current asthma. It was previously demonstrated that the presence of rhinoconjunctivitis can strongly increase the risk of current asthma [30]. As expected, at least in Western countries [31], increasing levels of AI produced a progressively increased asthma risk: nevertheless, it is noteworthy that acute respiratory diseases during the first two years of life produced a greater current asthma risk (OR= 8.53 [95%IC 4.15/17.52]) than the highest level of AI (OR = 3.03 [95%IC 1.37/6.70]), corresponding to individuals with at least three allergic sensitizations. In this model, males showed a lower current asthma risk than females (*p* = 0.046, Table 6). In fact, as age increases, asthma prevalence decreases in male subjects [32].

Despite the fact that allergic sensitization was more frequent among males and that male sex was protective against current asthma (Table 6), we did not find sex-related differences in prevalence rates of current asthma or rhinoconjunctivitis. This could be due to the known changes in asthma prevalence between females and males during puberty [33]. Thus, in our opinion, the cited disproportion between airways and air spaces present at birth between male and female babies continues producing its effects during childhood in terms of different responses to previous exposures (maternal smoking during pregnancy) and to atopy (allergic sensitization and Rhinoconjunctivitis), adding their effects to the sex-related differences in asthma over the lifespan [33].

### 4.4. Lung Function

We chose to use FEF_25–75%_ as a sensitive indicator of airway obstruction also in asymptomatic children [34]. In a multiple linear regression for FEF_25–75%_, we found that the occurrence of acute respiratory diseases during the first two years of life was correlated to poorer lung function. Males exposed to maternal smoking during pregnancy presented a highly negative B value (−0.375 [95% CI −0.596/−0.154], *p* = 0.001, Table 7) with respect to the combination of no exposure to maternal smoking during pregnancy and male sex. The remaining two combinations showed intermediate–but still significant–negative B values. This result suggests a sex-dependent sensitivity to maternal smoking during pregnancy exposure: in fact, when stratified by sex, the same multiple regression model provided different results. Acute respiratory diseases during the first two years of life significantly decreased FEF_25–75%_ during childhood in both females and males. In addition, only among males was FEF_25–75%_ also dependent on maternal smoking during pregnancy (Appendix A). This latter result was not confirmed for females, in agreement with previous reports indicating that maternal smoking during pregnancy has an effect on child lung function, but only in males [35]. In our sample, male children showed lower FEF_25–75%_/FVC ratio values than females, indicating that some inequalities between the geometry of the tracheobronchial tree and lung parenchyma may exist in male children with respect to females [36]. This seems to be confirmed by the results of the multiple regression for FEF_25–75%_/FVC ratio: exposure to maternal smoking during pregnancy negatively influenced males but not females. Finally, by evaluating “healthy” children only (i.e., children without respiratory symptoms), we found that exposure to maternal smoking during pregnancy and acute respiratory diseases during the first two years of life maintained their negative effect on lung function, even in those without any history of lower airway disease. This is in agreement with previous results showing that prenatal exposure to maternal smoking during pregnancy [35] and early-life lower respiratory tract infections [37] are associated with poorer lung function later in life.

### 4.5. Strengths and Limitations

The main strength of the present work is its comprehensive and simultaneous evaluation of many possible factors that, from in utero life to adolescence, may affect allergic sensitization and allergic diseases.

This work also presents some limitations. The reports relevant to parental smoking exposure during pregnancy are retrospective, thus they may be subject to recall bias. A more serious weakness is the lack of information on pregnancy duration, birthweight, and breastfeeding duration, which are known to significantly influence the outcome variables evaluated in this study. Nevertheless, it has been suggested that birthweight and gestational age may not be associated with lung function later in life [38]. However, the associations regarding sex-related differences and pre- and postnatal factors were obtained by means of an objective evaluation of individual allergic sensitization and lung function. Thus, these may provide useful additional insights on childhood asthma determinants. A further limitation of this study is the parental report of wheeze, which may be subject to misclassification. However, the core questions used represent the standard ISAAC measure for respiratory outcomes and have been shown to be predictive of doctor diagnosed asthma [11].

## 5. Conclusions

We confirm that acute respiratory diseases during the first two years of life–which appear to be influenced by a parental history of asthma–strongly influence both asthma and poorer lung function during childhood. Infections may compromise epithelial integrity, especially during early lung maturation phases, in turn facilitating sensitization to aeroallergens [39]. Thus, it is difficult to discriminate among all the complex interactions underlying the development of allergic respiratory diseases, and conditioning lung function, including both preventable and genetic factors (Figure 2). In particular, by means of a logistic association model stratified for parental history of asthma, we found that the prevalence of acute respiratory diseases during the first two years of life and exposure to maternal smoking during pregnancy–along with male sex–were significantly associated with allergic sensitization during childhood in the absence of a family history of allergic diseases. This strongly suggests that, without a genetic tendency to develop allergic diseases (i.e., the lack of parental history of asthma), exposure to maternal smoking during pregnancy and the occurrence of acute episodes of respiratory diseases in early life may act as strong independent risk factors for allergic sensitization. This, in turn, produces increased risk for allergic respiratory diseases.

## Figures and Tables

**Figure 1 ijerph-17-05087-f001:**
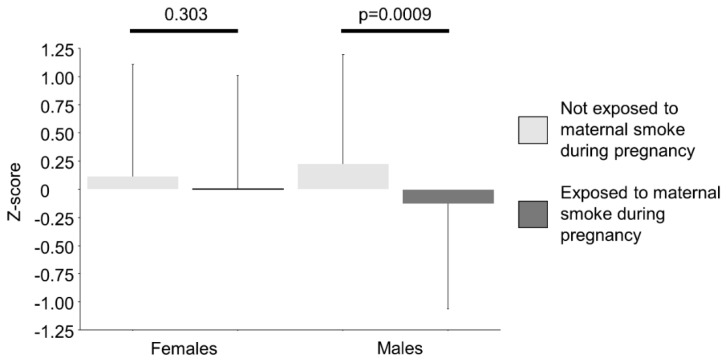
Means (± SD) of FEF_25–75%_ (z-score) separately by sex and exposure to maternal smoke during pregnancy. Differences were evaluated by means of one-way ANOVA and Fisher’s LSD.

**Figure 2 ijerph-17-05087-f002:**
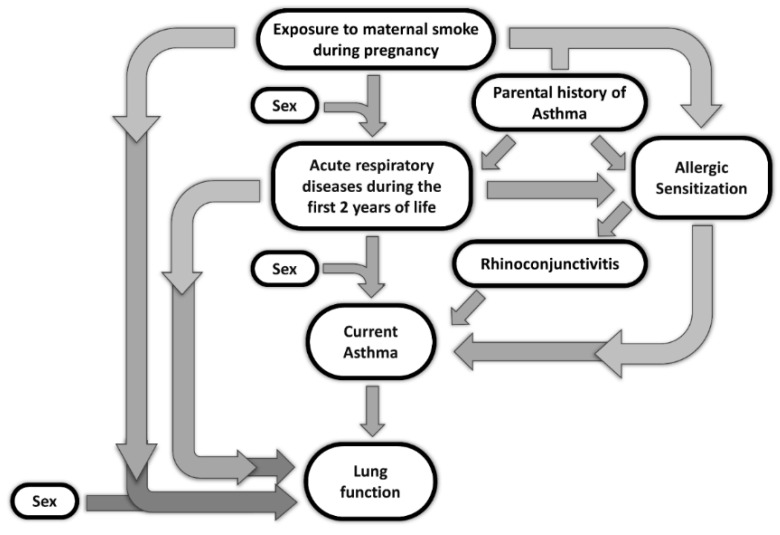
Schematic representation of the complex interactions underlying the development of acute respiratory diseases during the first two years of life and allergic respiratory diseases, as well as conditioning lung function, including both preventable and genetic factors.

**Table 1 ijerph-17-05087-t001:** Characteristics of study population. Significant *p*-Values are in bold.

	Overall Sample (N = 1714)	Without Current Asthma (N = 1661)	With Current Asthma (N = 53)	*p*-Value ^†^
Males (%)	51.4	51.7	43.4	0.23 **
Age (mean ± SD)	11.6 ± 1.4	11.6 ± 1.4	11.8 ± 1.1	0.38 *
Height, m (mean ± SD)	1.51 ± 0.12	1.51 ± 0.12	1.54 ± 0.09	0.17 *
Weight, kg (mean ± SD)	48.8 ± 13.5	48.1 ± 13.4	51.6 ± 14.2	0.12 *
BMI, kg/m^2^ (mean ± SD)	21.1 ± 5.4	21.0 ± 5.4	21.6 ± 4.2	0.50 *
FEV_1_ (z-score, mean ± SD)	0.10 ± 1.05	0.11 ± 1.04	−0.22 ± 1.14	**0.02** *
FVC (z-score, mean ± SD)	−0.24 ± 1.03	−0.24 ± 1.03	−0.31 ± 1.14	0.62 *
FEV_1_/FVC (%, mean ± SD)	90.4 ± 5.8	90.4 ± 5.8	88.0 ± 7.5	**0.003** *
FEF_25–75%_ (z-score, mean ± SD)	0.14 ± 0.99	0.16 ± 0.98	−0.46 ± 1.04	**<0.0001** *
FEF_25–75%_/FVC (mean ± SD)	1.12 ± 0.26	1.12 ± 0.26	0.99 ± 0.25	**0.0002** *
Exposed to maternal smoke during pregnancy (%)	12.1	12.0	15.1	0.50 **
Exposed to paternal smoke during pregnancy (%)	40.3	40.3	41.5	0.85 **
Exposure to environmental tobacco smoke during the first two years of life (%)	24.7	24.5	32.1	0.21 **
Acute respiratory diseases during the first two years of life (%)	31.9	24.5	32.1	0.21 **
Positive parental history of asthma (%)	12.3	11.5	35.8	**<0.0001** **
Current exposure to tobacco smoke at home (%)	46.8	46.5	54.7	0.24 **
Presence of mould/dampness at home (%)	10.5	10.5	11.5	0.81 **
Pet ownership (%)	21.9	21.8	24.5	0.63 **
Presence of heavy vehicular traffic in proximity to home (%)	23.6	23.6	22.6	0.87 **
Wheeze in the last 12 months (%)	6.3	3.3	100.0	**<0.0001** **
Doctor diagnosed asthma (%)	9.6	6.6	100.0	**<0.0001** **
Rhinoconjunctivitis (%)	11.1	10.2	41.5	**<0.0001** **
Allergic sensitization (%)	41.0	40.0	71.7	**<0.0001** **
No sensitization (%)	59.0	60.0	28.3	**<0.0001** **
One positive skin prick test (%)	17.0	16.7	24.5	
Two positive skin prick tests (%)	10.5	10.4	13.2	
Three or more positive skin prick tests (%)	13.5	12.9	34.0	

^†^*p*-Value computed for the differences between subjects with and without current asthma; * one-way ANOVA; ** χ^2^ test.

**Table 2 ijerph-17-05087-t002:** Multiple logistic association model for acute respiratory diseases in the first two years in the overall study population (*n* = 1714). OR: Odds Ratio. Significant *p* values are in bold.

Independent Variables	OR	95% Confidence Interval		*p*-Value
		Lower	Upper	
**Education level (Ref.: higher)**	0.821	0.660	1.022	0.078
Sex (Ref: Female)	1.282	1.038	1.593	**0.0214**
Exposure to maternal smoke during pregnancy (Ref.: not exposed)	1.792	1.295	2.479	**0.0004**
Paternal smoke during pregnancy (Ref.: not exposed)	0.874	0.680	1.122	0.291
Early exposure to environmental tobacco smoke (Ref.: not exposed)	1.167	0.879	1.548	0.286
Positive parental history of asthma (Ref.: negative)	1.516	1.113	2.065	**0.008**

**Table 3 ijerph-17-05087-t003:** Multiple logistic association model for acute respiratory diseases in the first two years in the overall study population. In this table, four combinations of exposure to maternal smoke during pregnancy and sex were separately evaluated. OR: Odds Ratio. Significant *p* values are in bold.

Independent Variables	OR	95% Confidence Interval		*p*-Value
		Lower	Upper	
**Education level (Ref.: higher)**	0.822	0.661	1.023	0.079
Paternal smoke during pregnancy (Ref.: not exposed)	0.869	0.677	1.117	0.273
Early exposure to environmental tobacco smoke (Ref.: not exposed)	1.174	0.884	1.559	0.267
Positive parental history of asthma (Ref.: negative)	1.508	1.107	2.055	**0.009**
Not exposed to maternal smoke during pregnancy * male sex ^†^	1.342	1.065	1.690	**0.013**
Exposed to maternal smoke during pregnancy * male sex ^†^	2.061	1.308	3.249	**0.002**
Exposed to maternal smoke during pregnancy * female sex ^†^	2.088	1.335	3.263	**0.001**

^†^ Ref.: Not exposed to maternal smoke during pregnancy * female sex.

**Table 4 ijerph-17-05087-t004:** Multiple logistic association model for allergic sensitization in the overall study population (*n* = 1714). OR: Odds Ratio. Significant *p*-Values are in bold.

Independent Variables	OR	95% Confidence Interval		*p*-Value
		Lower	Upper	
**Education level (Ref.: higher)**	1.047	0.847	1.295	0.669
Sex (Ref.: females)	1.483	1.207	1.822	**0.0002**
Positive parental history of asthma (Ref.: negative)	1.629	1.199	2.215	**0.002**
Maternal smoke during pregnancy (Ref.: not exposed)	1.378	0.989	1.920	0.058
Paternal smoke during pregnancy (Ref.: not exposed)	1.150	0.886	1.493	0.294
Early exposure to environmental tobacco smoke (Ref.: not exposed)	1.031	0.783	1.359	0.826
Acute respiratory diseases in the first two years (Ref.: not reported)	1.292	1.036	1.611	**0.023**
Current exposure to environmental tobacco smoke (Ref.: not reported)	0.921	0.718	1.181	0.515
Residence in proximity to roads with intense vehicular traffic (Ref.: not reported)	0.957	0.749	1.223	0.723
Mould/dampness at home (Ref.: not reported)	0.899	0.643	1.256	0.533
Pet ownership (Ref.: not reported)	0.923	0.718	1.186	0.532

**Table 5 ijerph-17-05087-t005:** Multiple logistic association model for allergic sensitization, separately for subjects with and without parental history of asthma. OR: Odds Ratio; 95% CI: 95% confidence interval. Significant *p*-Values are in bold.

	Subjects with Parental History of Asthma (N = 210)				Subjects without Parental History of Asthma (N = 1504)			
Independent variables ^†^	OR	95% CI		*p*-Value	OR	95% CI		*p*-Value
		Lower	Upper			Lower	Lower	
Education level	1.007	0.544	1.861	0.984	1.053	0.839	1.322	0.653
Sex	1.158	0.643	2.083	0.625	1.527	1.225	1.905	**0.0002**
Maternal smoke during pregnancy	0.673	0.268	1.689	0.399	1.537	1.076	2.194	**0.018**
Paternal smoke during pregnancy	1.282	0.649	2.530	0.474	1.137	0.855	1.511	0.376
Early exposure to environmental tobacco smoke	1.065	0.489	2.316	0.875	1.024	0.761	1.378	0.874
Acute respiratory diseases in the first two years	1.022	0.562	1.857	0.944	1.353	1.066	1.718	**0.013**
Current exposure to environmental tobacco smoke	1.066	0.546	2.083	0.851	0.915	0.698	1.198	0.518
Residence in proximity to roads with intense vehicular traffic	0.883	0.443	1.758	0.723	0.976	0.750	1.271	0.859
Mould/dampness at home	1.446	0.549	3.808	0.455	0.842	0.587	1.209	0.352
Pet ownership	1.138	0.583	2.220	0.705	0.890	0.677	1.169	0.402

^†^ Refer to Table 4 for reference levels.

**Table 6 ijerph-17-05087-t006:** Multiple logistic association model for current asthma in the overall study population (*n* = 1714). OR: Odds Ratio. Significant *p*-Values are in bold.

Independent Variables	OR	95% Confidence Interval		*p*-Value
		Lower	Upper	
**Education level (Ref.: higher)**	1.042	0.561	1.932	0.897
Sex (Ref.: Female)	0.531	0.286	0.988	**0.046**
Maternal smoke during pregnancy (Ref.: not exposed)	0.900	0.380	2.132	0.811
Paternal smoke during pregnancy (Ref.: not exposed)	0.797	0.371	1.713	0.561
Early exposure to environmental tobacco smoke (Ref.: not exposed)	1.490	0.683	3.251	0.317
Acute respiratory diseases in the first two years (Ref.: not reported)	8.526	4.150	17.516	**<0.0001**
Rhinoconjunctivitis (Ref.: not reported)	3.590	1.868	6.901	**0.0001**
Current exposure to environmental tobacco smoke (Ref.: not reported)	1.191	0.571	2.484	0.642
Residence in proximity to roads with intense vehicular traffic (Ref.: not reported)	0.933	0.449	1.936	0.852
Mould/dampness at home (Ref.: not reported)	0.810	0.315	2.085	0.662
Pet ownership (Ref.: not reported)	0.930	0.454	1.903	0.842
One positive skin prick test (Ref.: no sensitization)	2.283	1.010	5.159	**0.047**
Two positive skin prick tests (Ref.: no sensitization)	2.041	0.762	5.465	0.156
Three or more positive skin prick tests (Ref.: no sensitization)	3.025	1.367	6.695	**0.006**
BMI (kg/m^2^)	0.995	0.933	1.061	0.876

**Table 7 ijerph-17-05087-t007:** Multiple linear regression model for FEF_25–75%_ (as z-score) in the overall study population (*n* = 1714). Four combinations of exposure to maternal smoke during pregnancy and sex were separately evaluated. Significant *p*-Values are in bold.

Independent Variables	B	95% Confidence Interval		*p* Value
		Lower	Upper	
**Education level (Ref.: higher)**	−0.009	−0.111	0.092	0.856
Early exposure to environmental tobacco smoke (Ref.: not exposed)	0.098	−0.025	0.222	0.118
Acute respiratory diseases in the first two years (Ref.: not reported)	−0.270	−0.379	−0.161	**<0.0001**
Nocturnal cough (Ref.: not reported)	0.017	−0.107	0.141	0.792
Current asthma (Ref.: not reported)	−0.431	−0.733	−0.130	**0.005**
Exercise wheezing (Ref.: nor reported)	−0.081	−0.343	0.181	0.545
Rhinoconjunctivitis (Ref.: nor reported)	−0.124	−0.287	0.039	0.135
Current exposure to environmental tobacco smoke (Ref.: not reported)	0.064	−0.045	0.172	0.249
Residence in proximity to roads with intense vehicular traffic (Ref.: not reported)	0.080	−0.038	0.198	0.183
Mould/dampness at home (Ref.: not reported)	0.033	−0.127	0.193	0.685
Pet ownership (Ref.: not reported)	0.032	−0.087	0.151	0.595
Allergic sensitization (Ref.: negative)	0.089	−0.014	0.191	0.089
Weight (kg)	0.004	0.000	0.008	**0.032**
Not exposed to maternal smoke during pregnancy * female sex ^†^	−0.103	0.208	0.002	0.054
Exposed to maternal smoke during pregnancy * female sex ^†^	−0.243	−0.460	−0.026	**0.028**
Exposed to maternal smoke during pregnancy * male sex ^†^	−0.375	−0.596	−0.154	**0.001**

^†^ Ref.: not exposed to maternal smoke during pregnancy * male sex.

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
