# Peer review of "Determinants of Allergic Sensitization, Asthma and Lung Function: Results from a Cross-Sectional Study in Italian Schoolchildren"

_ijerph, 2020, doi:10.3390/ijerph17145087_

Round 1
Reviewer 1 Report
The aim of this paper was to evaluate prenatal, post-natal and early life risk factors for the presence of asthma in children aged 7-16 years in Italy. It provides information, about the complex relationships between prenatal & early life exposures, respiratory disease and sex with allergic sensitisation, asthma and reduced pulmonary function.
The tables were used well and clearly illustrate the results. Table 1 is simple and provides the results of the univariate analysis in a concise manner in addition to providing useful information about group sizes for various variables.
This paper uses a lot of non-standard abbreviations which makes reading it more tedious than necessary. I would suggest reconsidering the use of some of these non-standard abbreviations as I think reduction of the number of abbreviations will improve flow and readability.
The first paragraph of the discussion and conclusion paragraph provide good overviews of the study findings and bring the information together very well. Additionally, figure 2 is a good representation of the complex interaction of the risk factors for the development of allergic disease and asthma. However, there are multiple sentences in the discussion and the conclusion section that are very long, detracting from the readability. The manuscript will benefit from restructuring these.
Title:
The title “Determinants of Asthma in Italian Schoolchildren” does not wholly reflect the aims and results of this study and could be improved by simply adding a few words as per the first paragraph of the discussion
Language:
Minor grammatical and sentence structure adjustments required
Introduction:
Line 32-33: “in Italy a stabilizing trend of asthma prevalence has been observed” – this should be expressed more specifically. What does ‘a stabilizing trend’ mean? Trends usually infer change, but stabilising implies the opposite. Is the prevalence high or is it low and stable for x years or is it increasing or decreasing?
Materials & Methods:
Were the data assessed for normality? If so how? Please include this information as it is important to know & for you readers to decide whether your statistical approach is sound/valid
Line 123-124: “taking into account confounders and effect modifying variables” – how did you take this into account? Providing a little more detail about this process will add value & improved confidence in your statistical analysis.
Did you do a power analysis? Is your population size large enough considering the large number of factors/variables studied? Please provide details
Results:
Table 2, 4 & 5: providing an n-value and stating whether the model was done on the entire population (as per table 3) would improve clarity, consistency and will allow tables to be evaluated independently of text
Line 144: stating and thereby emphasising an OR of 1.54 with greater than 1 CI, seems unnecessary given the small magnitude of the OR – it feels like you are overstating the importance
Line 169-170 & Table 5 – “Ref.: not exposed to maternal smoke during pregnancy * male sex.” Why did you not use the same ref when evaluating sex and exposure to maternal smoking here as you did in Table 2 (ARD2Y multiple logistic association model – “Not exposed to maternal smoke during pregnancy * female sex.”) It seems counter intuitive given the repeated association with male sex to use male sex as the ref, but rather female sex should have been used as the reference as you did for the ARD2Y multiple logistic association model?
Table 5 – adding ‘combinations of exposure to maternal smoke during pregnancy and sex were evaluated’ to your table legend, as you did for table 2B, would improve clarity and allow it to be evaluated independently of the text
Line 174-176: why did you not also mention maternal smoke during pregnancy as the magnitude of the negative association was similar to that of ARD2Y
Line 180 & TableS3 – see previous comment for line 169-170 & Table 5
Discussion:
Line 193: “in utero exposure to maternal smoke during pregnancy …. ” – suggest to change to exposure to maternal smoke during pregnancy only, having both is superfluous and it is then consistent with the wording in your tables
Line 200-201: “Nevertheless, males exposed to MatSmoke showed twice the risk (OR 2.06) of having ARD2Y with respect to unexposed females” – this sentence is deceiving if a reader does not have time to thoroughly read & analyse the paper as infact both males & females have similar risk (with females OR slightly larger) of having ARD2Y – this sentence should reflect these results more accurately.
Line 211-217: Allergic sensitisation – there is a big difference in sample size of those with (210) vs without (1504) parental history of asthma. Given that your study results are different from that of other studies, it is worthwhile acknowledging that this smaller sample size may potentially have contributed to your results.
Line 219-221: “Similarly to what was observed for MatSmoke, subjects with ARD2Y not reporting ParAsthma presented a 45% prevalence of AS, while the same figure was 37% for children without ARD2Y (p=0.007)” – your sentence structure should be changed to improve clarity as in its current form, it is not clear/easily apparent that the sentence refers to children without ParAsthma either with or without ARD2Y. Additionally, this information is better suited to the results section and then discussed here rather than being presented in the discussion.
Line 229-232: The association with AI is not discussed in light of the literature & should be and strange that your AI-2 cases did not have a significant association with current asthma – any thoughts why?
Line 232-233: “In this model, males showed a lower CA risk than females (p=0.046, Table 4).” – given all your other results, the previously mentioned literature and your study’s title this finding warrants some expanded discussion
Line 241-242: “While ARD2Y significantly decreased FEF25-75% at childhood in both females and males, among males FEF25-75% also resulted dependent on atopy (i.e., AS and RC) and MatSmoke” – the meaning of the second half of this sentence is unclear and the sentence structure should be improved. In addition the claim of AS & RC being associated with reduced FEF in males is not supported by any of the results provided in the manuscript or the supporting document (RC p=0.157 & AS p=0.064). Please amend!
Line 253-258: this paragraph is better suited to 4.3 current asthma paragraph
Line 254: “despite the fact that male sex resulted protective for CA (Table 4)” – reconsider sentence structure
Line 262-263: “The reports relevant to parental smoking exposure during
pregnancy are retrospective” – 5 of the questions in the questionnaire including the parental smoking (as stated) and the ARD2Y question are historic questions (asking parents about factors that occurred at least 5 to 7 years prior) and are therefore highly likely to be subject to recall bias – this should be acknowledged more clearly in your limitation section
Line 265-268: This is a very long sentence. Consider rewording/restructuring to improve clarity and readability of this section.
Author Response
Manuscript ID: IJERPH-855532
Type of manuscript: Article
Title: Determinants of allergic sensitization, asthma and lung function: results from a cross-sectional study in Italian schoolchildren
Reviewer 1
The aim of this paper was to evaluate prenatal, post-natal and early life risk factors for the presence of asthma in children aged 7-16 years in Italy. It provides information, about the complex relationships between prenatal & early life exposures, respiratory disease and sex with allergic sensitisation, asthma and reduced pulmonary function.
Q: The tables were used well and clearly illustrate the results. Table 1 is simple and provides the results of the univariate analysis in a concise manner in addition to providing useful information about group sizes for various variables.
R: We thank the Reviewer for his/her appreciation of the manuscript.
Q: This paper uses a lot of non-standard abbreviations which makes reading it more tedious than necessary. I would suggest reconsidering the use of some of these non-standard abbreviations as I think reduction of the number of abbreviations will improve flow and readability.
R: According to the Reviewer’s suggestion, we have transformed abbreviations into the correspondent full text variables throughout the text. Consequently, the List of abbreviations has been deleted.
Q: The first paragraph of the discussion and conclusion paragraph provide good overviews of the study findings and bring the information together very well. Additionally, figure 2 is a good representation of the complex interaction of the risk factors for the development of allergic disease and asthma. However, there are multiple sentences in the discussion and the conclusion section that are very long, detracting from the readability. The manuscript will benefit from restructuring these.
R: The Discussion and Conclusion sections have been substantially rephrased.
Title:
Q: The title “Determinants of Asthma in Italian Schoolchildren” does not wholly reflect the aims and results of this study and could be improved by simply adding a few words as per the first paragraph of the discussion
R: As suggested, the manuscript title has been changed into Determinants of allergic sensitization, asthma and lung function: results from a cross-sectional study in Italian schoolchildren.
Language:
Q: Minor grammatical and sentence structure adjustments required
R: The manuscript was submitted to a thorough revision of English language.
Introduction:
Q: Line 32-33: “in Italy a stabilizing trend of asthma prevalence has been observed” – this should be expressed more specifically. What does ‘a stabilizing trend’ mean? Trends usually infer change, but stabilizing implies the opposite. Is the prevalence high or is it low and stable for x years or is it increasing or decreasing?
R: The first paragraph of the Introduction section has been rephrased.
Materials & Methods:
Q:Were the data assessed for normality? If so how? Please include this information as it is important to know & for you readers to decide whether your statistical approach is sound/valid
R: Normality of variable distribution was evaluated using the Kolmogorov-Smirnov Test: this is now stated at the beginning of the 2.5 “Statistics” paragraph.
Q: Line 123-124: “taking into account confounders and effect modifying variables” – how did you take this into account? Providing a little more detail about this process will add value & improved confidence in your statistical analysis.
R: It is well known that the relationships between genetic factors (parental history for asthma) and asthma are influenced by many behavioral (smoke exposure) and environmental (outdoor and indoor) conditions. Thus, in the multiple models, we introduced all the available variables relevant to possible outdoor (residence in proximity to roads with intense vehicular traffic [1]) and indoor (early and current exposure to environmental tobacco smoke [2], mould/dampness exposure at home [3], pet ownership [4]) exposures that were previously shown to produce an effect on the evaluated outcome variables. This information is now presented in paragraph 2.5 Statistics.
Q: Did you do a power analysis? Is your population size large enough considering the large number of factors/variables studied? Please provide details
R: Due to the design of the study, we did not perform a power analysis. Our population sample, that we believe is adequate for the statistical approach we adopted, includes 1,714 subjects. The multiple regression models we developed include a maximum of 14 independent variables: the only critical issue might be related to the analysis of subjects who reported Parental history of asthma, due to their limited number (N=210): this is now acknowledged in the “4.2. Allergic sensitization” paragraph, as requested.
Results:
Q: Table 2, 4 & 5: providing an n-value and stating whether the model was done on the entire population (as per table 3) would improve clarity, consistency and will allow tables to be evaluated independently of text
R: A statement that the analyses refer to the overall study population (n=1,714) has been introduced in the table legends.
Q: Line 144: stating and thereby emphasizing an OR of 1.54 with greater than 1 CI, seems unnecessary given the small magnitude of the OR – it feels like you are overstating the importance
R: As requested, the sentence has been removed.
Line 169-170 & Table 5 – “Ref.: not exposed to maternal smoke during pregnancy * male sex.” Why did you not use the same ref when evaluating sex and exposure to maternal smoking here as you did in Table 2 (ARD2Y multiple logistic association model – “Not exposed to maternal smoke during pregnancy * female sex.”)? It seems counter intuitive given the repeated association with male sex to use male sex as the ref, but rather female sex should have been used as the reference as you did for the ARD2Y multiple logistic association model?
R: In multiple logistic (Table 2) and multiple linear (Table 5) regression models, in evaluating the different combinations of sex and exposure to maternal smoking during pregnancy, we selected the combination (“Not exposed to maternal smoke during pregnancy * female sex” in Table 2B and “Not exposed to maternal smoke during pregnancy * male sex” in Table 5) producing all detrimental coefficients (i.e., positive in Table 2 and negative in Table 5B). Choosing a different reference would produce both positive and negative coefficients, causing – in our opinion – a less clear message.
Q: Table 5 – adding ‘combinations of exposure to maternal smoke during pregnancy and sex were evaluated’ to your table legend, as you did for table 2B, would improve clarity and allow it to be evaluated independently of the text
R: As requested, the sentence has been introduced in the Table 5 legend.
Q: Line 174-176: why did you not also mention maternal smoke during pregnancy as the magnitude of the negative association was similar to that of ARD2Y
R: The correlation between FEF25-75% and maternal smoke during pregnancy is now cited.
Q: Line 180 & TableS3 – see previous comment for line 169-170 & Table 5
R: The Reviewer is right: Table S3 has been corrected and the last paragraph of the Results section changed accordingly.
Discussion:
Q: Line 193: “in utero exposure to maternal smoke during pregnancy …. ” – suggest to change to exposure to maternal smoke during pregnancy only, having both is superfluous and it is then consistent with the wording in your tables
R: The sentence has been corrected accordingly.
Q: Line 200-201: “Nevertheless, males exposed to MatSmoke showed twice the risk (OR 2.06) of having ARD2Y with respect to unexposed females” – this sentence is deceiving if a reader does not have time to thoroughly read & analyse the paper as infact both males & females have similar risk (with females OR slightly larger) of having ARD2Y – this sentence should reflect these results more accurately.
R: As suggested by the Reviewer, the sentence has been modified.
Q: Line 211-217: Allergic sensitisation – there is a big difference in sample size of those with (210) vs without (1504) parental history of asthma. Given that your study results are different from that of other studies, it is worthwhile acknowledging that this smaller sample size may potentially have contributed to your results.
R: Following the Reviewer’s request, this is now acknowledged at the end of section “4.2. Allergic sensitization” of the manuscript.
Q: Line 219-221: “Similarly to what was observed for MatSmoke, subjects with ARD2Y not reporting ParAsthma presented a 45% prevalence of AS, while the same figure was 37% for children without ARD2Y (p=0.007)” – your sentence structure should be changed to improve clarity as in its current form, it is not clear/easily apparent that the sentence refers to children without ParAsthma either with or without ARD2Y. Additionally, this information is better suited to the results section and then discussed here rather than being presented in the discussion.
R: As suggested, the sentence has been rephrased and details removed (being presented in Results).
Q: Line 229-232: The association with AI is not discussed in light of the literature & should be and strange that your AI-2 cases did not have a significant association with current asthma – any thoughts why?
R: A reference has been added that is relevant to the association between allergic sensitization and asthma prevalence. Regarding the lack of significance in the AI-2 subgroup, this was likely due to the low number of subjects in it. In fact, AI-2, with 10.6% (N=181), represented the lowest subgroup in the Atopic Index classification (AI-0: 59.0%; AI-1: 17.0%; AI-3: 13.5%).
Q: Line 232-233: “In this model, males showed a lower CA risk than females (p=0.046, Table 4).” – given all your other results, the previously mentioned literature and your study’s title this finding warrants some expanded discussion
R: In the present work we did not find a significant difference in frequency distribution of sex among subjects with and without Current Asthma. Nevertheless, in the multiple logistic model for Current Asthma, male sex showed a “protective” OR (0.531 [0.286/0.988]) with respect to females. Because we moved – as per the Reviewer’s request – a paragraph from the Lung Function section to the Current Asthma section, we hope that this provides an adequate answer to the Reviewer’s request.
Q: Line 241-242: “While ARD2Y significantly decreased FEF25-75% at childhood in both females and males, among males FEF25-75% also resulted dependent on atopy (i.e., AS and RC) and MatSmoke” – the meaning of the second half of this sentence is unclear and the sentence structure should be improved. In addition the claim of AS & RC being associated with reduced FEF in males is not supported by any of the results provided in the manuscript or the supporting document (RC p=0.157 & AS p=0.064). Please amend!
R: We have tried to improve the clarity of the sentence and have performed the requested deletion.
Q: Line 253-258: this paragraph is better suited to 4.3 current asthma paragraph
R: As requested, the cited paragraph has been moved to the end of the Current Asthma paragraph and its structure has been changed (as per the next comment).
Q: Line 254: “despite the fact that male sex resulted protective for CA (Table 4)” – reconsider sentence structure
R: The sentence structure has been changed.
Q: Line 262-263: “The reports relevant to parental smoking exposure during pregnancy are retrospective” – 5 of the questions in the questionnaire including the parental smoking (as stated) and the ARD2Y question are historic questions (asking parents about factors that occurred at least 5 to 7 years prior) and are therefore highly likely to be subject to recall bias – this should be acknowledged more clearly in your limitation section
R: The risk of recall bias is now clearly declared in the “4.5. Strengths and limitations” section.
Q: Line 265-268: This is a very long sentence. Consider rewording/restructuring to improve clarity and readability of this section.
R: The paragraph has been rephrased.
References
- Ciccone G, Forastiere F, Agabiti N, et al. Road traffic and adverse respiratory effects in children. SIDRIA Collaborative Group. Occup Environ Med. 1998;55(11):771-778. doi:10.1136/oem.55.11.771
- Hofhuis W, de Jongste JC, Merkus PJ. Adverse health effects of prenatal and postnatal tobacco smoke exposure on children. Arch Dis Child. 2003;88(12):1086-1090. doi:10.1136/adc.88.12.1086
- Simoni M, Lombardi E, Berti G, et al. Mould/dampness exposure at home is associated with respiratory disorders in Italian children and adolescents: the SIDRIA-2 Study. Occup Environ Med. 2005;62(9):616-622. doi:10.1136/oem.2004.018291
- Lombardi E, Simoni M, La Grutta S, et al. Effects of pet exposure in the first year of life on respiratory and allergic symptoms in 7-yr-old children. The SIDRIA-2 study. Pediatr Allergy Immunol. 2010;21(2 Pt 1):268-276. doi:10.1111/j.1399-3038.2009.00910.x

Reviewer 2 Report
In my opinion, the article has too many abbreviations that make it difficult to read. The study’s objectives have to be expressed more clearly and the relationship between exposure to maternal smoke during pregnancy and the incidence of respiratory infections should be better explained (you have described only one possible mechanism: “increased airway responsiveness”). Moreover, since the conclusions you have reached to, are already known from previous studies, we may suggest the use of the term “confirm” instead of “find” to describe the outcome of your paper.
The figure number 2 should be improved in graphics and content.
Author Response
Manuscript ID: IJERPH-855532
Type of manuscript: Article
Title: Determinants of allergic sensitization, asthma and lung function: results from a cross-sectional study in Italian schoolchildren
Reviewer 2
Q: In my opinion, the article has too many abbreviations that make it difficult to read. The study’s objectives have to be expressed more clearly and the relationship between exposure to maternal smoke during pregnancy and the incidence of respiratory infections should be better explained (you have described only one possible mechanism: “increased airway responsiveness”). Moreover, since the conclusions you have reached to, are already known from previous studies, we may suggest the use of the term “confirm” instead of “find” to describe the outcome of your paper.
R: Also following the request of Reviewer #1, all the abbreviations were transformed into the correspondent full text variables throughout the text. Consequently, the List of abbreviations has been deleted. The last paragraph of the Introduction section has been expanded. The “4.1. Acute respiratory diseases in the first two years of life” paragraph in the Discussion section has been completely rewritten with additional references. As requested, in the first row of the Conclusions section, “found” has been changed to “confirm.”
Q: The figure number 2 should be improved in graphics and content.
R: As requested, Figure 2 has been completely redone in both its graphics and content.
